# Factors Affecting the Nuclei in Newborn and Children

**DOI:** 10.3390/ijerph19074226

**Published:** 2022-04-01

**Authors:** Christos Arnaoutoglou, Anastasia Keivanidou, Georgios Dragoutsos, Ioannis Tentas, Soultana Meditskou, Paul Zarogoulidis, Dimitrios Matthaios, Chrysanthi Sardeli, Aris Ioannidis, Eleni Isidora Perdikouri, Andreas Giannopoulos

**Affiliations:** 1Department of Obstetrics & Gynecology, Papageorgiou Hospital, Aristotle University of Thessaloniki, 54124 Thessaloniki, Greece; arnaoutoglou7@gmail.com; 2Pediatric Department, Aristotle University of Thessaloniki, AHEPA General Hospital, 54124 Thessaloniki, Greece; fbozali@outlook.com (A.K.); mkaranikas@outlook.com (A.G.); 3Department of Obstetrics and Gynecology, Democritus University of Thrace, 69132 Komotini, Greece; aesebidis@outlook.com; 4Department of Obstetrics & Gynecology, General Hospital of Giannitsa, 58100 Giannitsa, Greece; nmarkou1981@outlook.com; 5Laboratory of Histology and Embryology, Medical School, Aristotle University of Thessaloniki, 54124 Thessaloniki, Greece; zeinepmemet@outlook.com; 6Pulmonary Department, General Clinic Euromedica, 54124 Thessaloniki, Greece; 7Oncology Department, General Hospital of Rhodes, 85100 Rhodes, Greece; dimalexpoli@yahoo.com; 8Department of Pharmacology & Clinical Pharmacology, Faculty of Health Sciences, School of Medicine, Aristotle University of Thessaloniki, 54124 Thessaloniki, Greece; sardeli@auth.gr; 9Department of Surgery, “Genesis” Private Clinic, 54124 Thessaloniki, Greece; ariioann@yahoo.gr; 10Oncology Department, General Hospital of Volos, 38446 Volos, Greece; eigrouse@yahoo.gr

**Keywords:** radiation, newborn, children, environment pollutants, biomarker, environmental exposure, genetic damage, micronucleus assay

## Abstract

It is known that children are more sensitive to the effects of medical treatments and environment than adults. Today there is limited information regarding the differences in genotoxic effects in children. The micronucleus assay is a method that is used to monitor genotoxicity, and it was validated several years before. Today there is international interest for exfoliated buccal cells. Most of the micronuclei studies in children have been performed with the analyses of lymphocytes. However, there is vast interest in using exfoliated cells from the oral cavity. The reason is that other type of cells are acquired non-invasively, this is an important issue in paediatric cohorts. Unfortunately a limitation of measuring micronuclei frequency is that it has been observed to be low in newborns and on the other hand there are a large number of patients and cell sample counts. It has been observed that radiation exposure and environmental pollutants increase the micronuclei frequency in newborn and children. Regarding the medical treatments, there is little data and several studies are needed to optimise the doses. There is the need to observe if there is a relationship between micronuclei in lymphocytes and exfoliated cells and to identify the baseline of the micronuclei levels. Moreover, we evaluate the changes in response to the toxic agents. Prospective cohorts studies will clarify the predictive value of micronuclei for cancer and chronic diseases for both children and adults. Novel molecular technologies will assist in the elucidation of different biological pathways and molecular mechanisms connected with the micronulcei levels in newborn and children.

## 1. Introduction

In the present review we will provide information regarding the micronucleus (MN) assay in lymphocytes. Moreover, additional information will be given for exfoliated epithelial cells of children with a timeline from birth to adolescence. Children have higher sensitivity to genotoxic agents compared to adults. Furthermore, the genetic damage in younger ages affects adulthood health outcomes [1,2,3]. In recent years there has been a large number of studies with novel biomarkers of exposure that affect children. Several of these studies include genotoxicity assessments and micronuclei measurements in children [4,5]. As previously reported, the MN assay is the method of choice for evaluation of genotoxicity in newborns and children because it is cost-effective and efficient. The main factors affecting the micronuclei assay in newborn and children are: (a) age, (b) tissue specificity, (c) sex, (d) health status, and (e) environmental exposure. Firstly we present the reflexion of significance of micronuclei in newborns and then in children. Secondly, we present the effect of age and sex in different cell types. Moreover, we present the use of the micronuclei assay for monitoring of environmental exposures in paediatric patients along with chronic diseases and/or at the same time are undergoing treatment. Furthermore, we present the relationship of micronuclei as biomarkers of genotoxicity with the genetic polymorphisms and additional gene expressions. Thirdly and finally, we make an attempt to firstly summarize current research in genotoxicity of newborn and children. The significance of micronuclei in newborns and children micronuclei is observed in human T lymphocytes, which are produced from spontaneous chromosome breakage. T lymphocytes accumulate micronuclei; cells with abnormalities disappear except in the case of stable mutations in stem cells. MN in T lymphocytes from adults provides predictive data for carcinogenesis for early genetic diseases [6]. In newborns, the micronuclei in lymphocytes of umbilical blood indicate genome instability. There are four important factors:(a)T lymphocytes have to be investigated by micronuclei assay in circulating peripheral blood for 6 months [7],(b)T lymphocyte response to phytohaemagglutinin has been observed to be less efficient [8],(c)it has been observed that the baseline of micronuclei frequency is low in newborns.

Their predictive value of micronuclei for chronic diseases, and childhood leukaemia, requires still large prospective studies. In order for these future studies to be complete, we should combine information from different studies on exposure like nutrition, health lifestyle, and exposure to environmental factors during pregnancy. We should investigate micronuclei frequencies and genetic/epigenetic effects in mother–child pairs with the addition of disease record. Effects of age and sex on micronuclei levels. The connection between age and sex on micronuclei levels in adult lymphocytes has been previously established. Women have 30% higher levels of micronuclei than men and micronuclei levels increase with age [9]. There are not enough data regarding micronuclei in exfoliated epithelial cells [10]. Based on a meta-analysis of micronuclei frequency in children (age range 0–18 years) and data gathered from a pooled analysis from previously published studies and the Human Micronucleus International Collaborative Study (HUMN) database, we estimated the connection between age and gender on micronuclei level in peripheral lymphocytes [5,11]. There were low levels of micronuclei measured in lymphocytes for both boys and girls [4]. However, it was observed that micronuclei frequency was low at birth and increased by 66% in children ranging from 1–4 years of age. Micronuclei were then observed to increase by up to 116% in children aged 15–18 years. Every year there is a 6.5% increase in micronuclei frequency for children aged from to 5–13 years [12]. However, regarding buccal epithelial cells in another study with children aged from 14 to 18 years, there was no difference in micronuclei levels between boys and girls [13]. There have been additional studies on this issue with other studies reporting that the mean micronuclaei levels in buccal cells for children from 0 to 6 years differed by 2-fold (first study 1.2 and second study 3.8). This is thought to be influenced by the environmental factors. Furthermore, in a study involving lymphocytes and buccal cells of children βετςεεν ages 4 to 15 and their mothers, a 30% higher micronuclei frequency was observed in both cell types for adults when compared to children [14,15,16]. However, in this study, no statistical difference was observed in connection to age. On the other hand, there was a study reported from Madrid on newborns reporting that the mean number of micronuclei in binucleated cells was lower by 3.9 than in their parents 6.5 (Mothers) and 6.1 (Fathers) [17]. In another study from Mexico including four mother–child groups, a correlation was observed between mothers and newborn lymphocyte micronuclei frequency. In these groups, the levels of micronuclei frequency were lower than in previously published studies [18]. In another study including children of various ages, it was observed that the major factor affecting the micronuclei frequency was the air pollution [19]. This report is in accordance with the current information available that children are more vulnerable to environmental exposures [3].

## 2. Micronuclei and Environment

It is known that micronuclei assay are used to study genome damage in several medical situations like: (a) after transplacental and (b) accidental industrial/technological overexposures. There are several known genotoxic factors like: (a) unknown mixtures of airborne nanoparticles, (b) environmental tobacco smoke, (c) oil and coal combustion emissions, and (d) food contaminants. Higher frequencies of micronuclei have been observed in children exposed to environmental pollutants when compared to referent values [5]. Children exposed to air pollution have from 30 to 130% increase in the mean micronuclei level in comparison to referent values [15,20,21]. An additional 30% increase in micronuclei frequencies has been observed in those children exposed to indoor environmental tobacco smoke [22]. Increased levels of micronuclei were observed in lymphocytes and buccal cells of children due to the low regional ozone levels [16]. Based on a 4-year surveillance that included children from 3 to 7 years of age who were exposed to airborne chemical industry mixtures like: nitrates, ozone, formaldehyde, solvents and dust, a higher micronuclei frequency was observed in buccal cells than control children; in addition, an increase in the levels of micronuclei was observed in lymphocytes and buccal cells [23]. Higher micronuclei frequency was observed in binucleated lymphocytes Czech children aged from 5 to 13 years living in a polluted region, as compared with others of the same age living in a rural area without environmental pollution [12]. Higher micronuclei frequencies have been also observed in older children, meaning that there is a connection between age and exposure. There was an increase in micronuclei by 240% in children living near a chemical disposal site [24]. A 730% increase in micronuclei frequency has been observed in children that were exposed to heavy metals [14]. A 630% increase in micronuclei frequency has been observed in children exposed to arsenic [25]. In a study published from Poland, there were elevated micronuclei frequencies in lymphocytes from 9-year-old children who had higher blood lead levels in comparison to referant values [26]. Fluorescence in situ hybridization (FISH) assay was used and showed that genome damage was caused primarily by an aneugen mechanism. Moreover, it has been observed that natural sources of ionising radiation are connected with elevated micronuclei frequencies in schoolchildren exposed high radon levels [27]. Until now, the largest studies with ionising radiation in children have been available from studies associated with the Chernobyl, a nuclear accident [28,29]. Those children were chronically exposed to radiation after the Chernobyl nuclear power plant accident in 1986 and there was an increase in micronuclei levels in comparison to normal values [30]. Moreover, the children of the cleaners had also significantly increased micronuclei levels. Furthermore, it was observed that children from Belarus had higher micronuclei frequencies than children from other nearby countries. It was observed that there was a correlation between the levels of 137Cs and the presence of thyroid tumors in Belarus children [31]. The micronuclei frequencies were elevated. Since then, the micronuclei assay has been used for biomonitoring of genetic damage in contaminated regions of Belarus [32]. There is definitely a connection between early life environmental exposure and genetic damage in children, disease, and treatment on micronuclei. There are several studies published regarding the level of genetic damage of children with cancer with or without a combination of chemotherapy and radiotherapy. In a published study, there was no chromosome damage by micronuclei assay in lymphocytes of children and adolescents with thyroid carcinoma after receiving 131I radio therapy [33]. Changes in gene expression were evaluated and most patients had altered expression levels of DNA repair. In two previously published studies that evaluated the local radiation, no long-term association between micronuclei frequency as a result of dental X-rays [34] or radiosynovectomy was found [35]; as seen in Table 1.

Children who received 131I application for thyroid cancer around the vicinity of Chernobyl had an increase in micronuclei frequency for at least 5 days after the treatment. A decrease was observed from 4 to 7 months. In any case, the pretreatment levels were never restored [36]. In another earlier study from Chernobyl including children, in different parts of the former Soviet Union the results indicated increased levels of micronuclei in children [37]. The same group performed research on genotoxic effects of radiotherapy and chemotherapy for thyroid cancer. Most of the patients were treated with 131I and lower micronuclei levels were observed post-treatment by x10 in lymphocytes when compared to 60Co radiotherapy [21,25,26,27,28,29]. It was observed that chemotherapy resulted in an increase in micronuclei frequencies in lymphocytes and exfoliated buccal cells in paediatric patients with acute lymphocytic leukaemia when compared to the levels before treatment and in other healthy patients [38,39]. There are two additional examples where long-term therapies with hydroxyurea for the treatment of sickle-cell anaemia and methylphenidine for the treatment of deficit/hyperactivity disorder present long term genotoxicity [40]. Moreover, it was observed that the elevated micronuclei frequency remained for at least 12 years after hydroxyurea exposure. Increased micronuclei frequency was initially reported along with increased cytogenetic damage in children treated with methylphenidine [41]. Again, other later studies did not confirm this elevate micronuclei frequency in lymphocytes and buccal cells [42], nor in prospective follow-up up to 12 months [43]. Micronuclei studies in children have been conducted to characterize the differences in cytogenetic damage, which is associated with disease itself. It has been observed that the buccal cytome and micronuclei frequency are significantly altered in Down’s syndrome [44]. In other published studies in this patient group, genomic instability in blood and oral mucosa was also observed [45,46,47]. There are other studies regarding ataxia-telangiectasia and anaemia [48] and inflammatory bowel disease, where elevated micronuclei frequency was observed [49]. Based on the previous findings, we should keep in mind the genotoxicity of the diseases and combine with this information the adverse effects of chemo- and radiotherapy. By combining these, we can minimise the doses of the treatment but also at the same time keep the clinical effect. Moreover, we should monitor the genotoxicity of new treatment forms and the combination of therapies because they usually have a synergistic effect.

## 3. Micronuclei and Biomarkers

Until now there has been limited information in children regarding the gene expression and the relationship between the frequency of micronuclei and genetic polymorphisms. In the study by Decordier et al. [50], the impact of oxidative stress induced by H_2_O_2_ and the effect on the micronuclei frequencies was investigated in newborn girls. In the same study, the DNA repair (hOGG1, XRCC1, XRCC3, and XPD) and folate metabolism (MTHFR) polymorphisms were investigated. Increased micronuclei frequencies were observed in children carrying XRCC1194 variant genotype. It has been observed that newborns and children carrying the variant XRCC3241genotype are at higher risk for the induction of micronuclei oxidative stress. In a recent study by Rossnerova et al. from Czech Republic, the micronuclei frequency in the peripheral lymphocytes of children diagnosed with asthma was investigated. In this region, a high level of air pollution has been observed, which is assessed by concentrations of PM2.5 paricles, carcinogenic polycyclic aromatic hydrocarbons, and benzo[a]pyrene (B[a]P). These are the highest concentrations observed in Europe. It has been reported that the micronuclei levels in asthmatic children from 6 to 15 years of age have an impact on the modifying impact of genetic polymorphisms of GSTM1, GSTT1, and EPHX1. In this report, the micronuclei frequency in binucleated lymphocytes in asthmatic children had similar measurements to control subjects. Regarding GSTM1-positive subjects, a higher micronuclei frequency was observed vs. control subjects. A significant difference was observed for GSTT1 vs. control subjects. Furthermore, in this report, plasma analysis investigated vitamins C, A, and E along with cotinine level in urine and parental smoking. A multivariate linear regression analysis was performed and indicated the effect on GSTM1 gene regulation. In smoking families, two other genes significantly affected the micronuclei frequency of the GSTT1 and EPHX1. The support role of GSTM1 and possibly GSTT1 and EPHX1 genes is very important as they can be used to indicate the DNA damage in polluted environments. We still need larger studies like the combined cohort analysed by the HUMN in adults but also in children [51]. Currently there are ongoing studies attempting to investigate the genetic damage in children by novel methodologies such as the etheno adducts in mother–newborn pairs [52] and microarrays [34]. There is a report where micronuclei frequencies and gene expression in the lymphocytes of children and adults are different in different polluted areas [53]. In this study, very little difference was observed at the transcriptome level between children and adults, however, there were differences observed in children regarding the functions in nucleosome and immune response. Summary for micronuclei frequencies in children: Currently, micronuclei assay is increasingly used in newborns and children because of its public health significance. We can monitor the environmental exposures such as air pollution, industrial toxicants, radiation, and of course the toxicity of several medical treatments. We can also use this method to assess genotoxicity associated with leukaemia and Down’s syndrome. There are also other chronic conditions such as (a) asthma and (b) inflammatory bowel disease. Additional information has been gathered when assessing the micronuclei frequencies such as the nutritional and nutritional status. Today, the micronuclei frequency has been measured in children in lymphocytes and buccal epithelium; however, it has rarely been measured in reticulocytes. Until now, we focused on collecting lymphocytes for analyses, but in recent years, we collected exfoliated cells from the oral cavity because they can be collected non-invasively. This is an important issue when conducting paediatric studies [54]. Current information, as previously presented, indicate that the baseline micronuclei levels in newborn are low compared to adult values. Until now there has been a lack of knowledge regarding the effects of age and sex on the micronuclei levels in lymphocytes and exfoliated cells during childhood and later on in adolescence. We need further studies with a large number of subjects. There is still a knowledge gap regarding the changes in response to environmental exposures like: (a) chemical genotoxicants, (b) ionizing radiation, and (c) medical treatments that can alter the genome. Several studies have added the measurement of cytome along with the micronuclei that allows more detailed assessment of cell toxicity [44,54]. In other studies, FISH analyses of micronuclei [12], flow cytometry [40], and automatic scoring of micronuclei in different cell types has been used [55,56]. Fortunately, due to novel ‘omic’ and additional novel molecular methodologies with micronuclei studies [12], we will be able to enlighten the molecular mechanisms and biological pathways of the micronuclei levels in children.

## 4. Conclusions and Future Directions

Unfortunately, the levels of micronuclei frequencies are difficult to understand in paediatric populations due to interactions between environment, age, and sex. There is also the dynamic growth of the individual and adaptation to the environment parameters. Growth is a dynamic situation where several parameters have a significant impact to genome damage, which is measured by the micronuclei assay. Other parameters such as nutritional status and daily nutrition have not been taken under consideration in several studies because the primary focus is usually only one factor that possibly affects the micronuclei measurements. Moreover, there are very few studies with information regarding the health status and infectious diseases, which are also important factors and might affect the micronuclei measurements. Furthermore, cell division rate in children is different than in adults. It has been observed that epithelial cells from the oral cavity migrate to the surface from the basal layer in 2–3 weeks in adults. On the other hand, this time is much less in children and for different cell types. It is very important to know this time in the children population because we need to collect the cells for toxicity investigation in the right moment. To date, it is difficult to define the real biological significance of micronuclei frequencies in newborns and children. We definitely need standardization of the protocol blood collections or other cell types collections, maybe in correlation with the toxic factor under investigation. We would like to have a world bank of data from different regions and improve the quality of our results by comparing the length of the damage of a toxic agent in different populations and in different regions. We need to make a predictive cancer model for newborn and children. Definitely we need studies from pregnancy to childhood to adulthood. Finally we need ‘omics’ and novel molecular technologies presents for investigation of micronuclei in different cell types (Figure 1, Figure 2 and Figure 3).

## Figures and Tables

**Figure 1 ijerph-19-04226-f001:**
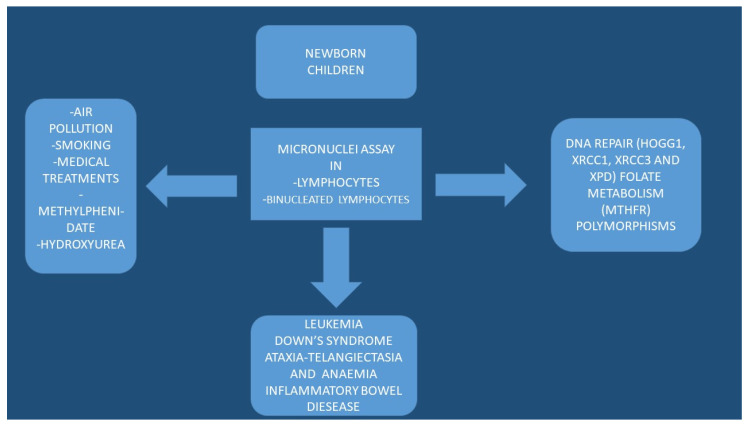
Graphical presentation of the factors affecting the nuclei and results.

**Figure 2 ijerph-19-04226-f002:**
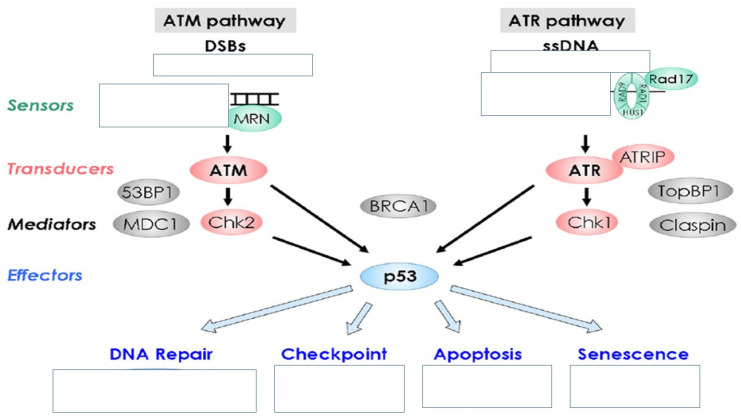
Damage to cellular DNA is involved in mutagenesis and the development of cancer. The DNA in a human cell undergoes a million damaging events per day, by both external (exogenous) and internal metabolic (endogenous) processes. Genomic mutations can also be carried over into daughter generations of cells if the mutation is not repaired prior to mitosis. Once cells lose their ability to effectively repair damaged DNA, there are three possible responses.

**Figure 3 ijerph-19-04226-f003:**
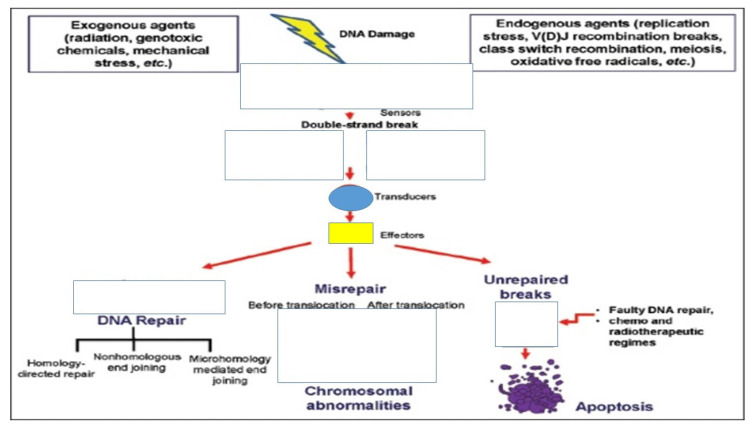
DNA damage response to a double strand break.

**Table 1 ijerph-19-04226-t001:** Micronuclei and environment.

(1)	Higher frequencies of micronuclei have been observed in children exposed to environmental pollutants
(2)	30% increase in micronuclei frequencies has been observed in children exposed to indoor tobacco smoke
(3)	There is a connection between age and exposure in the frequency of micronuclei
(4)	FISH—Fluorescence in situ hybridization is the best method to evaluate the micronuclei frequency
(5)	There is a connection between tumorigenesis and micronuclei frequency
(6)	Micronuclei frequencies can be used for biomonitoring of genetic damage
(7)	There is definitely a connection between early life environmental exposure and genetic damage in children.

## Data Availability

Not applicable.

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
