# Peer review of "Factors Affecting the Nuclei in Newborn and Children"

_ijerph, 2022, doi:10.3390/ijerph19074226_

Round 1
Reviewer 1 Report
The manuscript offers and important perspective. However, I would suggest sminor revisions to the manuscript:
- Improve keywords - I suggest including, biomarker, children, environmental exposure, genetic damage, micronucleus assay
- Table 1 is unconfigured in my version, I suggest editing

Author Response
Reviewer 1
The manuscript offers and important perspective. However, I would suggest sminor revisions to the manuscript:
- Improve keywords - I suggest including, biomarker, children, environmental exposure, genetic damage, micronucleus assay
Answer
Thank you for your comment
We have added the keywords that you suggested
- Table 1 is unconfigured in my version, I suggest editing
Answer
Thank you for your comment
We have made minor corrections to Table 1. We have added the most important information according to their importance

This manuscript is a resubmission of an earlier submission. The following is a list of the peer review reports and author responses from that submission.
Round 1
Reviewer 1 Report
The research topic is of great interest and may have an impact of early assessment and intervention. The authors introduce a large and importante discuss about micronucleus assay in newborns and young children. The review explain the scientific background and rationale for the investigation being reported and presents the main results. A minor review is suggested to better present the results on page 3, for example. There are repeated word ("another study" for introduce all study in this page ) and would be interesting to report these data in a table.

Author Response
Reviewer 1
The research topic is of great interest and may have an impact of early assessment and intervention. The authors introduce a large and importante discuss about micronucleus assay in newborns and young children. The review explain the scientific background and rationale for the investigation being reported and presents the main results. A minor review is suggested to better present the results on page 3, for example. There are repeated word ("another study" for introduce all study in this page ) and would be interesting to report these data in a table.
Answer
Thank you for your comments
Table 1 has been added according to your indication and linguistical corrections have been made in the section `Micronuclei and environment according to your indication`.

Reviewer 2 Report
This review presents the use of micronuclei assay to determine genotoxicity in pediatric patients and its value as a biomarker.
Good introduction, environment, biomarker, and conclusion sections have been included. While the review discusses the importance of micronuclei, the potential to use this as a biomarker is not described sufficiently. The authors can improve this review by including data to from published literature/previous studies, perhaps in a tabular form (for genetic polymorphisms, DNA damage, etc). Right now, this review reads as a book chapter and there is potential to improve this with a lit review, in vivo/in vitro experiments in graphical/tabular formats, and additional experiments that can be combined with this as a potential genotoxicity assay.
Author Response
Reviewer 2
This review presents the use of micronuclei assay to determine genotoxicity in pediatric patients and its value as a biomarker.
Good introduction, environment, biomarker, and conclusion sections have been included. While the review discusses the importance of micronuclei, the potential to use this as a biomarker is not described sufficiently. The authors can improve this review by including data to from published literature/previous studies, perhaps in a tabular form (for genetic polymorphisms, DNA damage, etc). Right now, this review reads as a book chapter and there is potential to improve this with a lit review, in vivo/in vitro experiments in graphical/tabular formats, and additional experiments that can be combined with this as a potential genotoxicity assay.
Answer
Thank you for your comments
We have added now Table 1, and made linguistical corrections in the section `Micronuclei and environment according to your indication`.
Moreover; we have added figure number 2 and 3 as you indicated that you wanted additional information regarding DNA damage.
We kept our review as a short review based on the invitation from the manuscript therefore we cannot add more information like in-vitro and in-vivo experiments as you indicate.

Round 2
Reviewer 2 Report
Authors have attempted to revise the manuscript based on reviewer comments.
Figured 1-3: Please make them schematically more graphical instead of words inside boxes. For example, apoptotic cell, the authors can draw a cell with signs of apoptosis. Making it more "graphical" with cartoons would help readers appreciate the review better.
Table 1 is not in the main manuscript. Please include.